# rPPG-HiBa:Hierarchical Balanced Framework for Remote Physiological Measurement

Submission Id: 2082

## ABSTRACT

Remote photoplethysmography (rPPG) is a promising technique for non-contact physiological signal measurement. It has great potential application in human health monitoring and emotion analysis. However, existing methods for the rPPG task ignore the long-tail phenomenon of physiological signal data, especially on multiple domains joint training. In addition, we find that the long-tail problem of the physiological label (phys-label) exists in different datasets, and the long-tail problem of domain exists under the same phys-label. To tackle these problems, in this paper, we propose a Hierarchical Balanced framework, which mitigates the bias caused by domain and phys-label imbalance. Specifically, we propose anti-spurious domain center learning tailored to learning domain-balanced embeddings space. Then, we adopt compact-aware continuity regularization to estimate phys-label-wise imbalances and construct continuity between embeddings. Extensive experiments demonstrate that our method outperforms the state-of-the-art in cross-dataset and intra-dataset settings.

## CCS CONCEPTS

• **Experience → Multimedia Applications**.

## KEYWORDS

physiological signal measurement, rPPG, imbalance, multimedia application

**ACM Reference Format:**

Anonymous Author(s). 2018. rPPG-HiBa:Hierarchical Balanced Framework for Remote Physiological Measurement. In *Proceedings of Make sure to enter the correct conference title from your rights confirmation emai (Conference acronym 'XX)*. ACM, New York, NY, USA, 9 pages. https://doi.org/XXXXXXX.XXXXXXX

## 1 INTRODUCTION

Remote photoplethysmography (rPPG) is a non-invasive technique that leverages video footage of the face to compute a wide range of physiological indicators. This technology has a variety of applications in multimedia, including health and fitness monitoring, emotion recognition, and biometrics [16, 21, 37, 38]. However, physiological signal data inherently exhibits non-uniform patterns. For

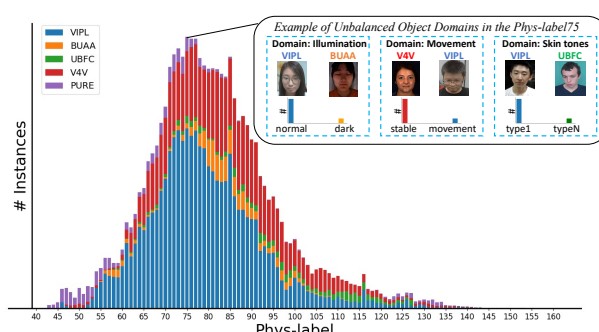

**Figure 1: The phys-labels (e.g., heart rate labels) suffer from a severe imbalance problem in VIPL, BUAA, UBFC, V4V, and PURE integrated datasets. Notably, there is also an imbalance problem in the distribution of sample domains (e.g., illumination, movement, and skin tones) under different phys-labels.**

instance, most heart rate data falls within the normal range of 60-95. Data that significantly deviates from this normal range often acts as an early indicator of potential health issues. Regrettably, instances of such atypical data are relatively rare. This imbalance in data distribution presents a significant challenge for rPPG-based measurement techniques [9, 19, 22, 26, 29, 32].

The traditional method [10, 17, 23, 28, 43, 44] for measuring physiological signals uses hand-crafted features. These algorithms ignore background noise, and it is difficult to obtain reliable results due to environmental variations. In recent years, deep learning (DL) networks have emerged as practical tools for rPPG tasks. These DL-based methods incorporate diverse input representations [13, 25, 26], network structures [6, 34, 47, 53], and loss constraints [7, 12, 33] to enhance model robustness. Notably, some studies [19, 20] utilize the integration of multiple datasets to improve the generalization performance of models.

However, all the existing methods ignore the imbalance problem on the rPPG task, and none of them consider that multi-dataset training further exacerbates this. The imbalance problem leads to the failure of the rPPG measurement task on minority samples. In this study, we conduct a comprehensive investigation of data imbalance issues in physiological signaling tasks, uncovering two distinct challenges:

*Domain Imbalance Challenge.* We refer to differences in attributes due to the diversity of environments, devices, subjects, etc., that the data are sampled from as domain differences in the data. We define the phenomenon that the number of data samples with the same phys-label in different domains varies significantly as domain imbalance. As illustrated in Fig 1, under the same phys-label, multiple datasets with different collection conditions are included. These datasets significantly differ in the number of instances, indicating

a domain imbalance among them. This phenomenon may lead to a bias in the representation of the tail domain towards the head domain during training, thus weakening the distinguishability of the representation of the tail domain. Consequently, the first challenge is addressing the domain imbalance problem to enhance the model's ability to represent samples well from the tail domain.

*Phys-label-wise Imbalance Challenge.* In addition to the domain imbalance, there is a significant imbalance among different phys-labels. As shown in Fig. 1, in the distribution of instances, a large number of instances are concentrated in a minority of phys-labels in the middle, and a small number of instances are distributed in a majority of phys-labels at both sides. Furthermore, differently from the traditional categorization problem, continuous phys-labels inherently convey meaningful distance information. However, the existing studies hardly exploit this property to construct continuity among embeddings under phys-label imbalance conditions. Therefore, the second challenge is addressing the imbalance among phys-labels to construct feature continuity among them.

To this end, we propose a Hierarchical Balanced framework that addresses the imbalance problem from both domain and phys-label-wise balanced perspectives, resulting a new method called **rPPG-HiBa** for short. To tackle with domain imbalance, we introduce an innovative approach called anti-spurious domain-centered learning (ADL). The method adopts an unsupervised clustering algorithm to estimate the domain distribution of the samples in each phys-label. It corrects biased feature centers by maintaining the feature centers of different domains, effectively improving representations' distinguishability and separability and mitigating the spurious correlation among embeddings caused by domain imbalance. To tackle with phys-label-wise imbalance, we propose a technique called compact-aware continuity regularization (CCR), which estimates the imbalance among physical labels by calculating the compactness of the embeddings in each phys-label's memorybank. In this way, categories with different degrees of imbalance can be reweighted more accurately. The continuity among embeddings can then be better constructed in the regression task.

This study intends to make the following new contributions to the field of rPPG:

- We are the first to systematically study the imbalance problem in rPPG tasks, and introduce a new challenge, the hierarchical imbalance problem.
- We develop two effective algorithms, ADL and CCR, for multi-level imbalance in the rPPG task. They alleviate the problem of domain imbalance without domain information and the inability to construct continuity among embeddings due to the hard-to-estimate degree of imbalance among phys-labels with continuity.
- The superiority of our approach is demonstrated by extensive cross-dataset and intra-dataset testing on five domain-rich and severely imbalanced datasets.

## 2 RELATED WORK

**Remote Physiological Measurement.** The goal is to estimate the HR and HRV values in a non-contact manner from the facial video captured by the camera. Traditional methods [10, 11, 17, 23, 28, 43–45] focus on converting facial video information into a continuously varying color space signal and separating the BVP signal utilizing signal processing. These methods are susceptible to external factors such as changes in the environment, light intensity, and slight movements, leading to the failure of the measurement task. In recent years, physiological signal measurement methods based on deep learning have shown excellent performance, with some methods [26, 39, 54, 54] improving the accuracy of measurements and speeding up inference by designing subtle model structures, some methods [20, 27, 47] making the extracted features more discriminative by designing strong feature constraints, and some methods [6, 19, 38] focusing on the noise brought by the data itself and eliminating the noise to make valuable features more significant. However, all these works ignore the problem of data imbalance. In contrast, we study the impact of data imbalance on physiological signal measurement tasks.

**Imbalanced Learning.** Imbalanced learning aims to improve the performance under-sample balance evaluation. Some previous work on imbalanced identification [1, 3, 24] mitigates inter-class imbalance by oversampling the tail data or downsampling the head data and reweighting the loss function. For imbalanced regression methods, some previous methods [30, 35, 41] transform the regression problem into a classification problem, using the idea of classification imbalance to solve the imbalanced regression problem. The method [51] addresses the interactions among successive labels and recalibrates the weights of different phys-labels of samples. Although the method considers the interactions of samples between successive labels, the weights are static versions, which are computed before training, the representation learning process is complex during training, and the prior computed weights may be inaccurate. For imbalanced contrastive methods. [8] points out that supervised contrastive learning is not directly used for classifying imbalance problems, and it performs even worse than cross-entropy loss. Some methods [46, 52] that also learn by contrast addressed the regression problem. However, all regression-related contrast learning efforts ignore the imbalance among data. For the domain balancing approach, recent studies [15, 40, 50] point out that learning invariant with features or separating domain-specific knowledge to enhance positive migration of domains can alleviate the domain imbalance problem. Contrary to all these studies, we are interested in hierarchical imbalance in the regression task.

## 3 PROPOSED METHOD

In this paper, we propose a hierarchical balanced framework, which can solve the imbalance problem on both domain and phys-label perspectives, as shown in Fig. 2. We design Anti-Spurious Domain Center Learning (ADL) and Compact Awareness Continuous Regularization (CCR) to address domain and phys-label imbalances, respectively. Specifically, ADL maintains the centers of the head, medium, and tail domains in memorybanks for each phys-label, constructing a new unbiased domain center to alleviate spurious correlations caused by domain imbalance and acquire more discriminative embeddings. Meanwhile, CCR computes the compactness of the embeddings stored in the memorybanks, reweights each category according to the compactness to mitigate the phys-label-wise imbalance, and constructs the phys-label-wise embeddings continuity.

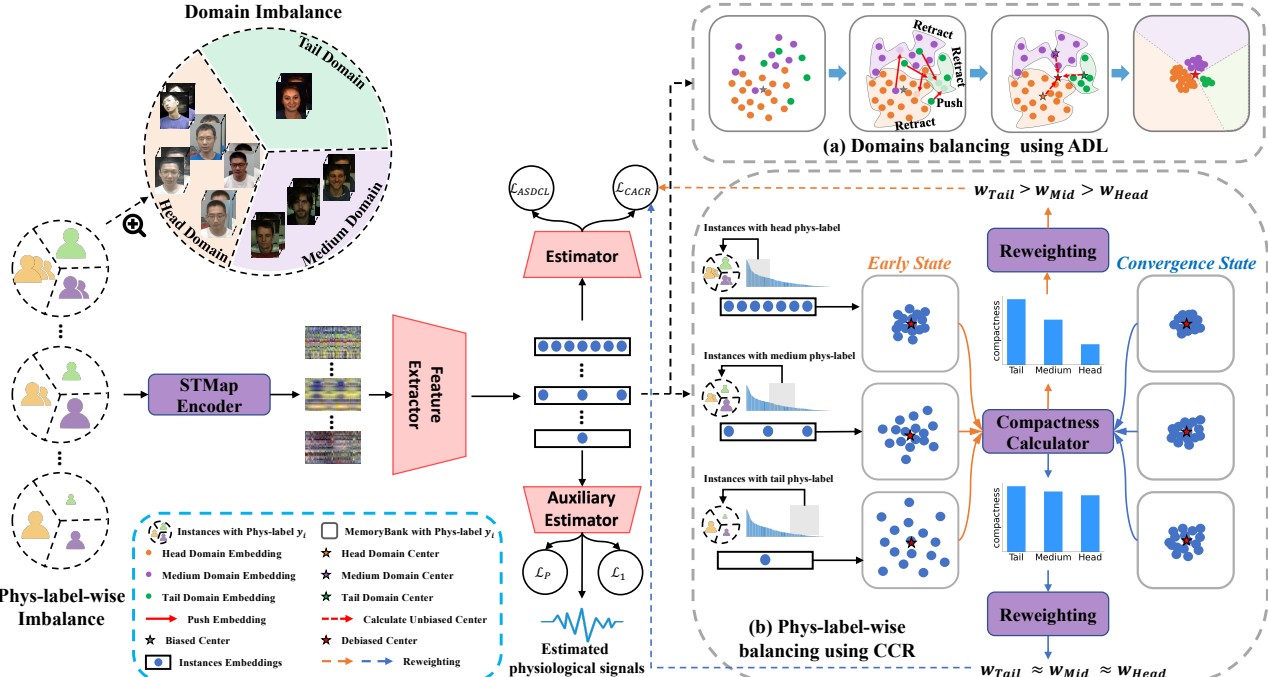

**Figure 2: The pipeline of the hierarchical balanced framework. Firstly, the input video sequence is transformed into a common representation, spatial-temporal map (STMap) [26]. Secondly, a shared feature extractor encodes all samples in the feature space. Thirdly, each phys-label maintains a memorybank of the same size, which stores the embeddings. Fourthly, the extracted instances' embeddings are inputted into an auxiliary estimator(the linear layer with dimension $d \times C$, where $d$ is the dimension of the embedding and $C$ is the number of physiologically labeled categories), where the embeddings are optimized using ADL(a) and CCR(b) algorithms. Finally, embeddings are regressed using the estimator(the linear layer with dimension $d \times 1$, where 1 represents a regression value), incorporating Pearson correlation coefficient loss and L1 loss. Introducing the auxiliary estimator during training aims to learn more discriminative and distinctive embeddings, while only the estimator outputs physiological predictions during testing.**

## 3.1 Domain Balance

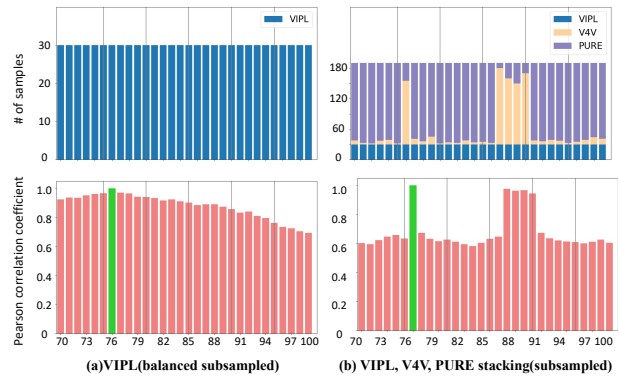

**Figure 3: (a) did not experience spurious correlation, whereas (b) did. The distribution of the training set data(top) and Pearson correlation coefficients(bottom) between the average embeddings at anchor on the test set and the average embeddings under other phys-labels.**

**Motivation Example.** As depicted in Fig. 3(a), in the VIPL dataset with phys-labels ranging from 70 to 100, 30 samples are uniformly sampled for each phys-label. We make a balanced VIPL

test set and computed the average embeddings for each phys-label on the test set. Phys-label 76 was selected as the anchor to calculate the Pearson correlation coefficient with the average embeddings of other phys-labels. Given the continuity of phys-labels, the closer the phys-label's average embeddings are to the anchor, the higher the Pearson correlation coefficient [51]. In Fig. 3(b), we uniformly sampled 190 instances for each phys-label, including VIPL, V4V, and PURE data. Among them, VIPL remained unchanged for training and testing, and we ensured that the sample quantity on the anchor point and phys-labels 87, 88, 89, and 90 of V4V data predominated, while VIPL was in the minority. Similarly, we tested using balanced VIPL. Suppose that the other two types of data and VIPL belong to the same domain. In that case, the expected Pearson correlation coefficient among the anchor and the average embeddings of other phys-labels should approximate the distribution in Fig. 3(a). However, we observed that the Pearson correlation coefficient among the anchor's average embedding and the average embeddings of phys-labels $87 \sim 90$, which are farther away, was higher. We can observe that these phys-labels are where the quantity of V4V predominates. This phenomenon arises because the domain of VIPL is notably different from that of V4V; for example, the lighting intensity of VIPL is significantly higher than that of V4V. VIPL's

domain is positioned at the tail on the anchor, weakening the discriminability among VIPL and V4V domain embeddings and leading to spurious correlation. To address this phenomenon, we should strive to enhance the discriminability and distinguishability of tail and head representations under the same phys-label, making the representations domain-independent.

*3.1.1 Compact Embedding Representation.* Naturally, the center loss can make the embeddings more compact within the the same phys-label, which can concurrently alleviate the problem of embedding overlap among adjacent phys-labels. The center loss function [48] is :.

$$\mathcal{L}_C = \frac{1}{2} \sum_{i=1}^{B} \| x_i - c_{y_i} \|_2^2 \qquad (1)$$

$c_{y_i} \in \mathbb{R}^d$ denotes the $y_i$th class center of deep embeddings. $B$ denotes the mini-batch size.

*3.1.2 Anti-Spurious Domain Center Learning.* As explained above, center loss exhibits favorable characteristics. However, the center loss does not account for domain imbalances within the phys-label. This can lead the model to incorrectly correlate some embeddings of the tail domain instances with the labels of the head domain instances. Consequently, the embedding center $c_{y_i}$ in Eq. 1 is more biased towards the head domain. This learning paradigm confuses embeddings in the tail and head domains, resulting in spurious correlations among embeddings.

For this reason, we develop Eq. 1 by extending it to a novel anti-spurious domain center loss, which can extract the domain-agnostic embeddings against domain imbalance. Specifically, the anti-spurious domain centers are obtained by calculating the mean value of the embedding centers maintained by the head, middle, and tail domain data, respectively. However, the data annotations do not contain accurate or missing domain labels. Fortunately, we can adopt an unsupervised approach to deduce whether each prototype in a class belongs to the head, middle, or tail domain based on its distance to the remaining embeddings. The distance formula is:

$$\psi(p_{y_i,j}) = \log \sum_{m=1}^{M-1} e^{s \cdot cos(e_{y_i,m}, p_{y_i,j})} \qquad (2)$$

$M - 1$ represents the number of embeddings in the memorybank except for the embedding prototypes. $s$ is a scaling hyperparameter. $e_{i,m}$ is the m-th embedding in the memorybank with phys-label $y_i$, $y_i \in \mathcal{Y}$, and $\mathcal{Y}$ is the set of de-duplication of all phys-labels. $p_{y_i,j}$ is the $j_{th}$ embedding prototype in the memorybank with phys-label $y_i$, $j \in \{1, 2, \cdots, M\}$. $cos(\cdot, \cdot)$ is a function measuring the distance between two embeddings. Ideally, if the embeddings are uniformly distributed, each embedding will have the same $\psi(p_{y_i,j})$. Otherwise, the embeddings with larger $\psi(p_{y_i,j})$ are more likely to come from a tail domain [4]. Hence, the embeddings can be classified into corresponding domains by clustering algorithms:

$$\mathcal{S}_{y_i,d} = f_c(\psi(p_{y_i,1}), \psi(p_{y_i,2}) \cdots, \psi(p_{y_i,M})) \qquad (3)$$

$f_c$ denotes a clustering algorithm, such as K-Means. $\mathcal{S}_{y_i,d}$ is the set of embedding distributions within phys-label $y_i$, where $d$ represents the domain embedding distribution, and $d \in \mathcal{D} := \{head, mid, tail\}$. $c_{y_i,d}^t$ is the updated embedding centers of the different distribution domains.

When obtaining the distribution of embedding domains within the phys-label $y_i$, a center $c_{y_i,d}^{t-1}$ is maintained for the head, middle, and tail domains at moment $t - 1$. Each center is updated as follows:

$$\triangle c_{y_i,d}^{t-1} = \frac{\sum_{i=1}^{B} \delta(x_i \in \mathcal{S}_{y_i,d}) \cdot (c_{y_i,d}^{t-1} - x_i)}{1 + \sum_{i=1}^{B} \delta(x_i \in \mathcal{S}_{y_i,d})} \qquad (4)$$

$$c_{y_i,d}^t = c_{y_i,d}^{t-1} - \alpha \cdot \triangle c_{y_i,d}^{t-1} \qquad (5)$$

where $\alpha$ is the hyperparameter, and $\delta(\cdot, \cdot)$ is the conditional function. The anti-spurious domain center $c_{y_i}^*$ is obtained by averaging the embedding centers of $n$ different distribution domains in the memorybank under phys-label $y_i$, as in the following equation:

$$c_{y_i}^* = \frac{1}{n} \sum_{d \in \mathcal{D}} c_{y_i,d}^t \qquad (6)$$

Anti-spurious domain centers can better separate different distributional domain embeddings to obtain a domain-balanced embedding space, thus alleviating the problem of spurious correlation among embeddings. Combined with compact discriminative representation learning, the loss of domain-distributed anti-spurious domain center learning can be expressed by the following equation:

$$\mathcal{L}_{ADL} = \frac{1}{2} \sum_{i=1}^{B} \| x_i - c_{y_i}^* \|_2^2 \qquad (7)$$

## 3.2 Phys-label-wise Balance

**Motivation Example.** Inspired by [51], we utilized two types of data: (1) the CIFAR100 dataset, which is a 100-class classification dataset from which we sampled a subset of data, and (2) VIPL data with continuous phys-labels. We sampled instances from VIPL that matched the label density distribution of CIFAR100 and had the same label range, set from 40 to 100 (Fig. 4), as there were almost no phys-labels below 40 in the heart rate data. We make both test sets balanced. Subsequently, we trained a regular ResNet-18 model on these two datasets and plotted their test error distributions. As depicted in Fig. 4(a), the error distribution exhibits a strong negative correlation with the label distribution. This phenomenon is expected because majority classes, having more instances, tend to perform better than minority classes in learning. However, from Fig. 4(b), we observe significant differences in error distribution for VIPL, which has a continuous label space, even though the label density distribution is the same as CIFAR-100. Notably, the error distribution appears smoother and is no longer closely correlated with the label density distribution. This phenomenon is because, in continuous scenarios, the empirical label distribution does not necessarily reflect the real label density distribution due to dependencies among data samples from neighboring labels (e.g., images with nearby physical labels). For this scenario, the method LDS [51] employs a symmetric kernel convolved with the empirical density distribution and considers the overlap of nearby label data sample information. It estimates a label distribution close to the true one, thereby addressing the imbalance issue by compensating for the real label density distribution.

However, the problem we need to address is more complex. Our data consists of multiple domains under the same phys-label. In contrast, the data addressed by LDS has only one domain under continuous labels and does not suffer from domain imbalance issues.

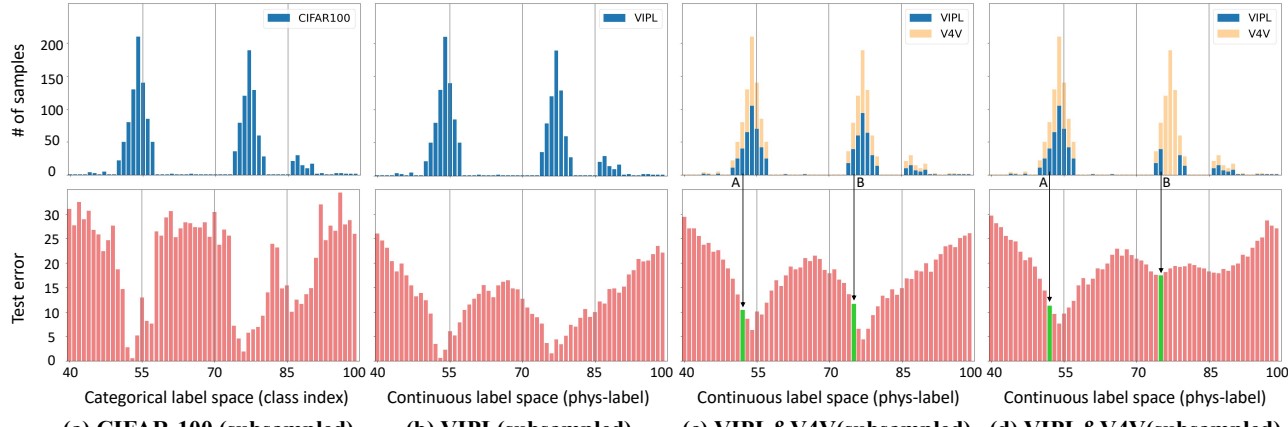

**Figure 4: Comparing test error distributions (bottom) across different datasets with the same training label distribution (top): (a) CIFAR-100, a classification task with a discrete label space. (b) VIPL is a regression task with a continuous label space. (c) V4V is also a regression task with a continuous label space, but instances in a distinct domain compared to VIPL, such as significantly different lighting conditions between VIPL and V4V. Here, V4V and VIPL instances are equally distributed within each phys-label. (d) VIPL has only one instance in labels 76-78; the rest are V4V instances. For other labels, V4V and VIPL instances are equally distributed.**

We conducted additional control experiments. We used VIPL and V4V data with continuous labels but different domains. In Experiment 1, the total amount of VIPL and V4V samples and the density distribution of labels were the same as those in Fig. 4(a) and Fig. 4(b), and VIPL and V4V each accounted for half of the samples at each phys-label. Testing was performed on balanced VIPL data. This setup aimed to simulate multi-domain scenarios with domain balance. As shown in Fig. 4(c), the error distribution is similar to that in Fig. 4(b), which aligns with intuition. Although the data quantity decreases, VIPL's data distribution approximates that in Fig. 4(b). In this scenario, LDS can still estimate the real label distribution. In Experiment 2, we make slight modifications to the data. The total amount of VIPL and V4V samples and the density distribution of labels were the same as those in Fig. 4(c), and VIPL and V4V each accounted for half of the samples at each phys-label (except for phys-labels 76 to 78, where only one VIPL data was present, and the rest were V4V). Testing was also performed on balanced VIPL data. This setup aimed to simulate multi-domain scenarios with domain imbalance. Interestingly, as observed in Fig. 4(d), the error distribution is no longer similar to that in Fig. 4(b) or Fig. 4(c). Compared to Fig. 4(c), the errors at point B in Fig. 4(d) (phys-label=75) are significantly higher, as the availability of data from the same domain (VIPL) on the right side of point B in Fig. 4(d) decreases, thereby altering the error distribution. The multi-domain and imbalanced domain scenarios make it difficult to estimate the real label distribution, making it challenging to compensate for phys-label-wise imbalance based on the real true label distribution. So, can we mitigate the phys-label-wise imbalance phenomenon from another perspective?

#### 3.2.1 Continuous Embedding Representation Learning.

In the rPPG measurement task, phys-labels have two essential properties: continuity and imbalance. When the distribution of continuity labels is relatively balanced, the work [55] reduces fragmentation of representation by capturing the continuity of regression tasks in

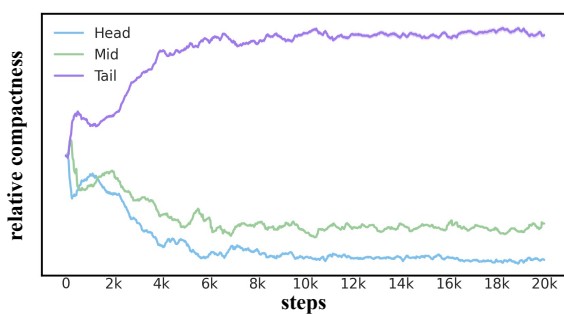

**Figure 5: The relative compactness of embeddings among head, middle, and tail phys-labels. The more clustered the embeddings are, the smaller the relative compactness value is.**

scenarios with continuity labels. And learning awareness regression representations by comparing distances among sample targets. Given a training mini-batch of $N$ input and phys-label pairs $\{(x_n, y_n)\}_{n \in [N]}$, we augment [20] the mini-batch as $\{(\tilde{x}_\ell, \tilde{y}_\ell)\}_{\ell \in [2N]}$, where $\tilde{x}_{2n}, \tilde{x}_{2n-1}$ are randomly augment using input $x_{2n}$, and $\tilde{y}_{2n} = \tilde{y}_{2n-1} = y_{2n}, \forall n \in [N]$. These augmented images are fed into the encoder, yielding embedding $v_\ell, \forall \ell \in [2N]$. Thanks to [55] for continuity embedding representation learning, we have:

$$\mathcal{L}_{CFRL} = -\frac{1}{2N} \sum_{i=1}^{2N} \frac{1}{2N-1} \sum_{j=1, j \neq i}^{2N} \log \frac{exp(v_i \cdot v_j / \tau)}{\sum_{k=1}^{2N} cond \cdot exp(v_i \cdot v_k / \tau)} \quad (8)$$

$$cond = \mathbb{1}_{[k \neq i, d(\tilde{y}_i, \tilde{y}_k) \geq d(\tilde{y}_i, \tilde{y}_j)]} \quad (9)$$

where $\tau$ is temperature parameter, $d(\cdot, \cdot)$ is $L1$ distance function, $\mathbb{1}_{[\cdot]} \in \{0, 1\}$ is a conditional function, which is 1 if the condition is met and 0 otherwise.

#### 3.2.2 Compact-Aware Continuity Regularization.

For the phys-label-wise imbalanced rPPG measurement task, mitigating the imbalance problem among labels when performing continuum embedding

representation learning is necessary to construct discriminative representations [8]. In addition, it is difficult for the label distribution to fully reflect the degree of imbalance in the data due to the complex dependency problem among consecutive labels. Thus, it is inaccurate to directly reweight the samples based only on the distribution of labels. Within phys-labels, we construct compact representations. In short, the representation of easy samples (e.g., head samples) will be more compact than that of hard samples (e.g., tail samples). We can see from Fig. 5 that the head embeddings were more compact in the experiment. Inspired by this, we can more accurately estimate the imbalance among phys-labels by calculating the compactness of the representations for re-weighting.

Calculating the distance among all sample embeddings in the phys-label and the central embedding is inefficient. In order to improve computational efficiency, we design a memorybank for each phys-label, which is used to store a certain amount ($M$=16 or $M$=32 or $M$=64) of embeddings. The degree of compactness ($\mu$) of all representations within each bank can then be calculated as follows:

$$\mu_{y_i} = \frac{\log \sum_{m=1}^{M} e^{s \cdot cos(e_{y_i,m}, c^*_{y_i})}}{\sum_{i=1}^{N} \log \sum_{m=1}^{M} e^{s \cdot cos(e_{y_i,m}, c^*_{y_i})}} \tag{10}$$

where $M$ represents the size of memorybank, $N$ represents the number of phys-labels, $e_{y_i,m}$ represents the m-th embedding in the $i$-th phys-label memorybank, $c^*_{y_i}$ is the center embedding of the $i$-th phys-label memorybank, the distance of two embedding $e_{y_i,m}$, $c^*_{y_i}$ is formulated as $s \cdot cos(e_{i,m}, c^*_{y_i})$, $s$ represents the expansion coefficient. We can dynamically estimate the corresponding weight w of the updated memorybank for each physical label based on the embedding compactness of the $i$-th phys-label, and the following equation gives the weight vector $\vec{w}$ composed of $w_{y_i}$:

$$\vec{w} := norm(\frac{1}{\mu_{y_1}} \sum_{i=1}^{N} \mu_{y_i}, \cdots, \frac{1}{\mu_{y_N}} \sum_{i=1}^{N} \mu_{y_i}) \tag{11}$$

$norm$ denotes the normalization operation. $\vec{w}$ is the normalized weights vector.

To build a learning process for labeling semantically continuous embeddings with imbalance, we introduce a compact-aware continuity regularization based on continuous embedding representation learning (Eq. 8). The regular term is then computed over the embeddings as:

$$\mathcal{L}_{CCR} = -\frac{1}{2N} \sum_{i=1}^{2N} \frac{1}{2N-1} \sum_{j=1,j\neq i}^{2N} \log \frac{w_{y_i} \cdot w_{y_j} \cdot exp(v_i \cdot v_j/\tau)}{\sum_{k=1}^{2N} cond \cdot w_{y_i} \cdot w_{y_j} \cdot exp(v_i \cdot v_k/\tau)} \tag{12}$$

$$cond = \mathbb{1}_{[k\neq i, d(\tilde{y}_i, \tilde{y}_k) \geq d(\tilde{y}_i, \tilde{y}_j)]} \tag{13}$$

$w_{y_i}$, $w_{y_j}$ are denoted as the weights of the embeddings labeled $y_i$, $y_j$, which are the $i$, $j$-th elements in $\vec{w}$.

CCR mitigates the imbalance problem among phys-labels by compactness awareness and constructs the continuity of embeddings. At a high level, it can increase the weight of hard samples (tail data or data that is difficult to rely on continuous data to supplement information on itself) when the head data has learned more discriminative embeddings than the tail data. When the hard samples have acquired more compact representations, increasing the head samples that have been slowed down for learning is appropriate. So on and so forth, both head and tail samples are fully learned.

## 3.3 Overall Loss

For the rPPG measurement task, it is necessary to construct the L1 loss $\mathcal{L}_{L1}$ between the target label and the predicted label. Also, constructing a negative Pearson correlation coefficient loss $\mathcal{L}_P$ for data with BVP signals can make the predicted physiological signal value distribution closer to the true value distribution. In addition, introducing $\mathcal{L}_{ADL}$ loss and $\mathcal{L}_{CCR}$ loss can make the embeddings more discriminative and generalizable. However, the L1 loss and P loss are first given more considerable weight in the early stage of training, which can accelerate the convergence of the model [14]. Therefore, we introduce the adaptation factor $\gamma = \frac{2}{1+exp(-10\cdot\hat{g})}$, $\hat{g} = \frac{iter_{current}}{iter_{total}}$, and the stabilization factor $k_1 - k_4$. The overall loss function is as follows:

$$\mathcal{L} = k_1 \cdot \mathcal{L}_{L1} + k_2 \cdot \mathcal{L}_P + \gamma \cdot (k_3 \cdot \mathcal{L}_{ADL} + k_4 \cdot \mathcal{L}_{CCR}) \tag{14}$$

## 4 EXPERIMENTS

### 4.1 Datasets and Implementation Details

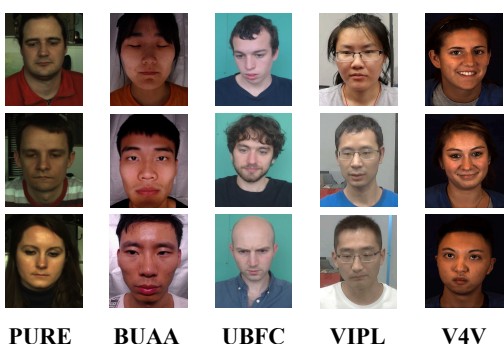

**PURE    BUAA    UBFC    VIPL    V4V**

Figure 6: Samples from five different datasets.

**Datasets.** We conduct our evaluation on five widely used datasets, which are PURE [36], BUAA [49], UBFC [2], VIPL [26], and V4V [31]. These datasets contain rich scenarios such as different lighting conditions, motions, skin tone types, etc. A more detailed description of these datasets can be found in **Appendix. B**.

**Implementation details.** Our framework is implemented based on pytorch. For the input processing of the model, the generation process of STMap is the same as that of [26]. For fair comparison, we utilize the same model ResNet-18, augmentation method, and evaluation method as [20]. The memorybank is a matrix of dimension $C \times M \times d$, where $C$ is the number of categories, $M$ is the size of the memorybank, and $d$ is the dimension of the stored embeddings. Memorybank can perform the insertion of new embeddings and the push out of old embeddings. The super parameters $k_1 - k_4$ are set as 1,1,0.01, 0.1. According to the scale of the loss function. $s$, $\tau$, $\alpha$, $M$ are set as 1, 0.07, 0.7, 64 respectively. The learning rate of Adam was set to 0.001, and batchsize was set to 2048 under **cross-dataset testing protocol**(Train on four source domains and test on another unknown domain. e.g. Train on V4V, BUAA, PURE, UBFC, test on VIPL), 512 under **intra-dataset testing protocol**(Train on VIPL, test on VIPL), and the total number of learning inters was set to 20000.

**Metric indicator.** The common indicator for evaluating HR estimation in physiological signal measurement are mean absolute error(MAE), root mean square error (RMSE), and Pearson's correlation (r). For the dataset with BVP signals, the Heart Rate variability (HRV) (i.e., Low Frequency (LF), High Frequency (HF), LF/HF, HR-BVP) is evaluated using MAE, RMSE, r.

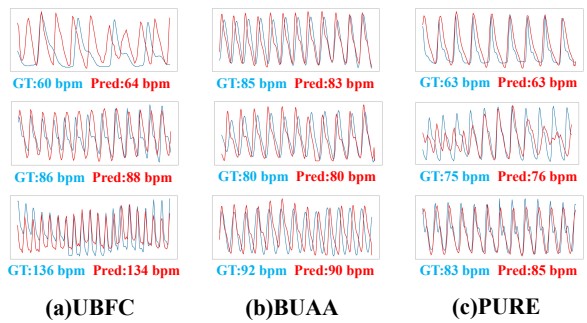

GT:60 bpm  Pred:64 bpm     GT:85 bpm  Pred:83 bpm     GT:63 bpm  Pred:63 bpm

GT:86 bpm  Pred:88 bpm     GT:80 bpm  Pred:80 bpm     GT:75 bpm  Pred:76 bpm

GT:136 bpm  Pred:134 bpm   GT:92 bpm  Pred:90 bpm     GT:83 bpm  Pred:85 bpm

**(a)UBFC          (b)BUAA          (c)PURE**

**Figure 7: Visualization of model predictions (Red) and ground truths (Blue) on cross-dataset testing on UBFC, BUAA, and PURE.**

## 4.2 Intra-dataset Testing

We selected VIPL data sets is the most challenging datasets for rPPG task, obviously, there is an the long tail problem. As shown in Tab. 1, our method was compared with many SOTA method including traditional measurement methods (SAMC, POS, CHROM) and DL-based measurement methods (I3D, DeepPhy, BVPNet, CVD, Physformer, Dual-GAN, NEST, Baseline (base on RhythNet)). These results are derived from [19, 20, 25]. rPPG-HiBa significantly improved baseline method and achieved optimal performance. This shows that the imbalance problems of both domain and labels are important problems that hinder model performance, and our approach can significantly solve this problem.

## 4.3 Cross-dataset Testing

*4.3.1 HR Estimation.* To further evaluate the generalization performance of our algorithm, we applied a more challenging cross-dataset testing protocol for heart rate estimation [20]. In addition to the existing rPPG method, we also reproduced two long-tail algorithms specifically designed for regression problems, namely LDS and BMSE. As illustrated in Tab. 2, our proposed method outperforms the existing methods on all datasets. These results highlight that the hierarchical long-tail problem poses a significant obstacle to the model's generalization performance, and our algorithm effectively mitigates this issue through the use of anti-spurious domain center and compact-aware continuity regularization. Moreover, as shown in Fig. 7, we randomly sampled video clips on the datasets(UBFC, BUAA, PURE) with BVP signals and visualized the rPPG estimates.

*4.3.2 Domain Balance Analysis on BUAA, VIPL.* We selected VIPL and BUAA, two datasets rich in domain-specific information, to assess the performance of our proposed algorithms across various domains. Previous research in physiological signal estimation has introduced tailored algorithms, including NEST, Dual-GAN and

DOHA, designed to address domain-specific challenges. NEST focuses on domain generalization, while the Dual-GAN and DOHA methods aim to reduce non-signal domain information. We conduct a comparative analysis of these two algorithms. The results are shown in Tab. 3 and Tab. 4. And we performed ablation experiments on the $\alpha$ and $M$ parameters in ADL, which can be found in **Appendix. C**. Significantly, our proposed method not only enhances but also balances the performance across each domain. More detailed metrics can be found in **Appendix. D**.

*4.3.3 Phys-label Balance Analysis on V4V, VIPL.* We chose two datasets with challenging data imbalances, V4V and VIPL. We compared our algorithms using NEST, a state-of-the-art estimation algorithm for physiological data, and LDS, a superior algorithm for data with continuity and imbalance. As can be seen from the comparison results of VIPL and V4V in Tab. 5, the NEST algorithm focuses more on the performance of the head data but adversely affects the performance of the tail data. On the contrary, the LDS algorithm performs well in the tail but poorly in the head. In contrast, our proposed CCR algorithm employs a dynamic reweighting approach that more accurately estimates the degree of imbalance among the data. It demonstrates excellent performance on both head and tail data. More detailed metrics can be found in **Appendix. E**. And we performed ablation experiments on the $\alpha$ and $M$ parameters in CCR, which can be found in **Appendix. C**

*4.3.4 HRV Estimation.* The HRV is used to evaluate the model's performance on data with BVP signals. We use two datasets for training (any two of UBFC, PURE, BUAA) and one other for testing fellowing [20]. As shown in **Appendix. F**, compared with the traditional methods GREEN, CHROM, POS, and Baseline, our proposed method has achieved the best evaluation on LF, HF, LF/HF, and HR-bmp indicators. The results show that alleviating the hierarchical long-tail problem is an efficient and direct way to improve the physiological signal measurement task.

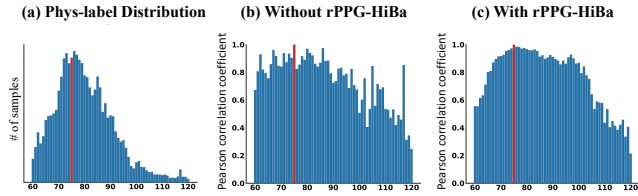

(a) Phys-label Distribution     (b) Without rPPG-HiBa     (c) With rPPG-HiBa

**Figure 8: Pearson correlation among embeddings with anchor HR75 and embeddings labeled HR60-HR120.**

*4.3.5 Embedding Correlation and Continuity Analysis.* For the regression task, embeddings with continuity can mitigate the fragmentation of the representation. We calculated the Pearson correlation coefficients with other HRs using HR75 as an anchor. As can be inferred from Fig. 8, the correlation among embeddings is smoother with the rPPG-HiBa framework. In addition, it can effectively mitigate the imbalance among labels and the a priori that the tail data directly inherits from the head data, thus mitigating the spurious correlation of the tail data.

| Method | SD↓ | MAE↓ | RMSE↓ | r↑ |
|---|---|---|---|---|
| **SAMC** [42] | 18.0 | 15.9 | 21.0 | 0.11 |
| **POS** [44] | 15.3 | 11.5 | 17.2 | 0.30 |
| **CHROM** [10] | 15.1 | 11.4 | 16.9 | 0.28 |
| **I3D** [5] | 15.9 | 12.0 | 15.9 | 0.07 |
| **DeepPhy** [6] | 13.6 | 11.0 | 13.8 | 0.11 |
| **BVPNet** [9] | 7.75 | 5.34 | 7.85 | 0.70 |
| **CVD** [27] | 7.92 | 5.02 | 7.97 | 0.79 |
| **Physformer** [54] | 7.74 | 4.97 | 7.79 | 0.78 |
| **Dual-GAN** [19] | 7.63 | 4.93 | 7.68 | 0.81 |
| **NEST** [20] | 7.49 | 4.76 | 7.51 | 0.84 |
| **DOHA** [38] | 7.69 | 4.95 | 7.73 | 0.80 |
| **Baseline** [26] | 8.04 | 5.21 | 8.07 | 0.77 |
| **rPPG-HiBa** w/o $\mathcal{L}_{CCR}$ | 7.57 | 4.83 | 7.61 | 0.82 |
| **rPPG-HiBa** w/o $\mathcal{L}_{ADL}$ | 7.33 | 4.54 | 7.34 | 0.84 |
| **rPPG-HiBa** | **7.26** | **4.47** | **7.28** | **0.85** |

Table 1: Comparison of the HR estimation results of our method with some other state-of-the-art methods on the VIPL-HR database. Bold indicates the best result of all compared methods. The symbol ↑ higher is better, and ↓ lower is better.

| Method | UBFC | | | PURE | | | BUAA | | | VIPL | | | V4V | | |
|---|---|---|---|---|---|---|---|---|---|---|---|---|---|---|---|
| | MAE↓ | RMSE↓ | r↑ | MAE↓ | RMSE↓ | r↑ | MAE↓ | RMSE↓ | r↑ | MAE↓ | RMSE↓ | r↑ | MAE↓ | RMSE↓ | r↑ |
| **GREEN** [43] | 8.02 | 9.18 | 0.36 | 10.32 | 14.27 | 0.52 | 5.82 | 7.99 | 0.56 | 12.18 | 18.23 | 0.25 | 15.64 | 21.43 | 0.06 |
| **CHROM** [10] | 7.23 | 8.92 | 0.51 | 9.79 | 12.76 | 0.37 | 6.09 | 8.29 | 0.51 | 11.44 | 16.97 | 0.28 | 14.92 | 19.22 | 0.08 |
| **POS** [44] | 7.35 | 8.04 | 0.49 | 9.82 | 13.44 | 0.34 | 5.04 | 7.12 | 0.63 | 14.59 | 21.26 | 0.19 | 17.65 | 23.22 | 0.04 |
| **DeepPhys** [6] | 7.82 | 8.42 | 0.54 | 9.34 | 12.56 | 0.55 | 4.78 | 6.74 | 0.69 | 12.56 | 19.13 | 0.14 | 14.52 | 19.11 | 0.14 |
| **TS-CAN** [18] | 7.63 | 8.25 | 0.55 | 9.12 | 12.38 | 0.57 | 4.84 | 6.89 | 0.68 | 12.34 | 18.94 | 0.16 | 14.77 | 19.96 | 0.12 |
| **Dual-GAN*** [19] | 5.55 | 7.62 | 0.79 | 7.24 | 10.27 | 0.78 | 3.41 | 5.23 | 0.84 | 8.88 | 11.69 | 0.50 | 10.04 | 14.44 | 0.35 |
| **BVPNet*** [9] | 5.43 | 7.71 | 0.80 | 7.23 | 10.25 | 0.78 | 3.69 | 5.48 | 0.81 | 8.45 | 11.64 | 0.51 | 10.01 | 14.35 | 0.36 |
| **NEST*** [20] | 4.77 | 7.03 | 0.85 | 6.89 | 9.98 | 0.84 | 2.67 | 3.89 | 0.90 | 7.57 | 10.87 | 0.60 | 9.87 | 13.53 | 0.40 |
| **DOHA** [38] | 4.83 | 7.19 | 0.83 | 6.97 | 10.03 | 0.84 | 3.05 | 4.62 | 0.87 | 7.89 | 11.13 | 0.56 | 9.92 | 13.48 | 0.38 |
| **LDS*+** [51] | 4.06 | 6.43 | 0.87 | 7.35 | 10.80 | 0.81 | 2.68 | 3.88 | 0.90 | 7.94 | 11.35 | 0.56 | 10.68 | 13.70 | 0.34 |
| **BMSE*+** [30] | 4.26 | 6.78 | 0.87 | 7.81 | 10.46 | 0.84 | 2.74 | 4.09 | 0.89 | 7.80 | 11.27 | 0.55 | 10.69 | 13.75 | 0.30 |
| **Baseline*** [26] | 5.79 | 7.91 | 0.78 | 7.39 | 10.49 | 0.77 | 3.38 | 5.17 | 0.84 | 8.97 | 12.16 | 0.49 | 10.16 | 14.57 | 0.34 |
| **rPPG-HiBa*+** w/o $\mathcal{L}_{CCR}$ | 4.00 | 7.10 | 0.85 | 7.19 | 11.28 | 0.82 | 2.84 | 3.74 | 0.96 | 7.67 | 10.60 | 0.60 | 9.72 | 13.29 | 0.39 |
| **rPPG-HiBa*+** w/o $\mathcal{L}_{ADL}$ | 3.93 | 6.87 | 0.86 | 6.79 | 9.90 | 0.85 | 2.76 | 3.63 | 0.96 | 7.51 | 10.58 | 0.60 | 9.72 | 12.84 | 0.40 |
| **rPPG-HiBa*+** | **3.75** | **6.60** | **0.88** | **6.43** | **9.44** | **0.87** | **2.45** | **3.28** | **0.98** | **7.34** | **10.41** | **0.61** | **9.69** | **12.54** | **0.42** |

Table 2: HR estimation results on cross-dataset testing protocol (Train on four source domains and test on another unknown domain. e.g. Train on V4V, BUAA, PURE, UBFC, test on VIPL). * means that STMap is used as input, and + means that the imbalanced method is applied to a baseline (baseline without GRU).

| Method | v1/s | v2/s | v3/s | v4/s | v5/s | v6/s | v7/s | v8/s | v9/s | s1/v | s2/v | s3/v | all |
|---|---|---|---|---|---|---|---|---|---|---|---|---|---|
| **Baseline*** [26] | 9.53 | 8.28 | 11.61 | 7.62 | 8.29 | 8.36 | 7.95 | 9.12 | 12.21 | 9.39 | 8.47 | 9.26 | 8.97 |
| **Dual-GAN** [19] | 8.81 | 9.16 | 9.74 | 8.70 | 7.78 | 8.11 | 9.29 | 8.63 | 10.87 | 8.84 | 8.74 | 9.12 | 8.88 |
| **DOHA*** [38] | 7.94 | 7.61 | 8.49 | 6.21 | 8.22 | 7.25 | 8.38 | 9.15 | 10.06 | 7.85 | 7.67 | 8.20 | 7.89 |
| **NEST*** [20] | **7.05** | 7.33 | 8.21 | **5.82** | 7.94 | **7.12** | 8.07 | 9.39 | 10.11 | 7.66 | **7.10** | 8.08 | 7.57 |
| **Baseline***+$\mathcal{L}_{ADL}$ | 7.82 | 7.52 | 8.37 | 7.00 | 7.61 | 7.58 | 7.37 | 7.64 | 8.93 | 7.80 | 7.50 | 7.78 | 7.67 |
| **rPPG-HiBa*+** | 7.20 | **7.26** | **7.81** | 6.34 | **7.73** | 7.32 | **7.29** | **7.33** | **8.68** | **7.31** | 7.33 | **7.39** | **7.34** |

Table 3: Cross-dataset testing results (MAE↓) on different domains on VIPL to validate domain balancing (+ADL). The dataset contains nine different scenarios (v1-v9), which are captured with three different devices (s1-s3). We consider the data of different acquisition devices for the same scenario as one domain, e.g., the v1/s domain represents all the data captured by the s1,s2, and s3 devices in the v1 scenario. The same acquisition devices for different scenarios are also considered as one domain, e.g., the s1/v domain, which represents all the data of v1-v9 captured by the s1 device. More detailed descriptions of the scenarios and devices can be found in Appendix. 2.

| Method | NI | MI | HI | all |
|---|---|---|---|---|
| **Baseline*** [26] | 3.84 | 3.30 | 3.02 | 3.38 |
| **Dual-GAN** [19] | 3.88 | 3.07 | 3.30 | 3.41 |
| **DOHA*** [38] | 3.29 | 2.94 | 3.09 | 3.05 |
| **NEST*** [20] | 3.11 | 2.45 | 2.47 | 2.67 |
| **Baseline***+$\mathcal{L}_{ADL}$ | 2.93 | 2.81 | 2.77 | 2.84 |
| **rPPG-HiBa*+** | **2.56** | **2.38** | **2.41** | **2.45** |

Table 4: Cross-dataset testing results (MAE↓) on different domains on BUAA to validate domain balancing (+ADL). The dataset contains rich "illumination" domain information. We set the samples with 10.0lux and 15.8lux as "normal illumination" (NI), the samples with 25.1lux and 39.8lux as "medium illumination" (MI), and the samples with 63.1 lux and 100 lux as "high illumination" (HI).

| Method | head >50 | medium ≤50 & ≥23 | tail <23 | overall |
|---|---|---|---|---|
| **Baseline*** [26] | 5.76 | 11.24 | 21.35 | 8.97 |
| **NEST*** [20] | **4.81**(+0.95) | 9.05(+2.19) | 19.21(+2.14) | 7.57(+1.40) |
| **LDS*+** [51] | 6.07(-0.31) | 8.52(+2.72) | **16.78**(+4.57) | 7.94(+1.03) |
| **Baseline***+$\mathcal{L}_{CCR}$ | 5.34(+0.42) | 8.35(+2.89) | 17.31(+4.04) | 7.51(+1.46) |
| **rPPG-HiBa*+** | 5.15(+0.61) | **8.33**(+2.91) | 16.95(+4.40) | **7.34**(+1.63) |

| Method | head >14 | medium ≤14 & ≥8 | tail <8 | overall |
|---|---|---|---|---|
| **Baseline*** [26] | 7.23 | 11.82 | 20.02 | 10.16 |
| **NEST*** [20] | **6.46**(+0.77) | 12.17(-0.35) | 20.70(-0.68) | 9.87(+0.29) |
| **LDS*+** [51] | 8.48(-1.25) | 11.34(+0.48) | 19.12(+0.90) | 10.68(-0.52) |
| **Baseline***+$\mathcal{L}_{CCR}$ | 6.97(+0.26) | **11.17**(+0.65) | 19.09(+0.93) | 9.72(+0.44) |
| **rPPG-HiBa*+** | 6.90(+0.33) | 11.29(+0.53) | **18.98**(+1.04) | 9.69(+0.47) |

Table 5: Cross-dataset testing on VIPL(left) and V4V(right) datasets to validate phys-label-wise balancing (+CCR). Results (MAE↓) of different methods on three disjoint subsets. For VIPL, We classify phys-labels with more than 50 samples as head, fewer than 23 as tail, and the remaining as medium. For V4V, We classify phys-labels with more than 14 samples as head, fewer than 8 as tail, and the remaining as medium. Green indicate how the method performance exceeds the baseline. Red indicate how the method performs inferior to the baseline.

## 5 CONCLUSION

The imbalanced distribution of physiological signal data poses a significant challenge to remote physiological measurement based on rPPG in multimedia applications. This study proposes a novel hierarchical balanced framework to address this issue. For the domain imbalance, the framework obtains a domain-balanced embedding space by learning the anti-spurious domain center. For the phys-label-wise imbalance, the framework mitigates the imbalance problem among labels by using compactness awareness and constructs the continuity of embeddings. Exhaustive experimental results demonstrate that alleviating the hierarchical imbalance enhances the generalization and discriminative of the model, thereby greatly advancing the state of the art in the area of rPPG.

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
