# OpenReview forum: "rPPG-HiBa:Hierarchical Balanced Framework for Remote Physiological Measurement"
_acmmm.org/ACMMM/2024/Conference — MM2024 Poster_

### Official Review · Reviewer_bJTX · 2024-05-04

**Rating:** 3
**Confidence:** 4

**Summary:**

The authors introduce a Hierarchical Balanced framework to mitigate the bias caused by domain and phys-label imbalance for enhancing rPPG estimation. In particular, the authors introduce anti-spurious domain center learning tailored to learning domain-balanced embeddings space. In addition, the authors adopt compactaware continuity regularization to construct continuity within latent rPPG feature space.

**Strengths:**

The authors conduct comprehensive experiments to evaluate their proposed method.

**Limitations:**

1 The framework in Figure 2 is very unclear and confusing.

1.1	What are L_ ASDCL and L_CACR?  "L_ASDCL" and "L_CACR" are not mentioned in the main text.

1.2	In the caption of Figure 2 , the author say that `` Introducing the auxiliary estimator during training aims to learn more discriminative and distinctive embeddings, while only the estimator outputs physiological predictions during testing. ‘’ What are the estimator and the auxiliary estimator?

1.3	Why use L_1 and L_P to constrain the auxiliary estimator and use L_ ASDCL and L_CACR to constrain the estimator? If only using L_ASDCL and L_CACR alone, it's unclear how the estimator would learn rPPG estimation.

2	The novelty of this paper is incremental.
The proposed Anti-Spurious Domain Center Learning is not a new idea and highly similar to [A] (Please refer to Figure 4 of [A]). Next, in the proposed Continuous Embedding Representation Learning, the idea of reweighting is a common approach in addressing long-tail problems.

3	The overall writing is confusing and difficult to read, and there may be some errors in the equations.

3.1	In Eq. (13), it is unclear why positive weights (w_yj) are used in the denominator of contrastive learning.

3.2	Confusing symbols.

3.2.1 	In Eq. (1), it is unclear what x_i represents, as the author has not explained it in this section. However, in Section 3.2.1 Continuous Embedding Representation Learning, we see that the author represents x_n as the input video. It is confusing why the input video can be operated on with latent features in Eq. (1).

 3.2.2Inconsistent symbols. The authors adopt various symbols, such as x in Eq. (1), e and v, to represent the latent features. The differences among x, e, and v are not clearly explained.

3.2.3 What is alpha in the Section C of Appendix?   alpha is not mentioned in the main text.

4   In the proposed Anti-Spurious Domain Center Learning, why is K-means capable of discerning the features of the tail domain? When setting K=3, it only ensures the separation of three distinct features, but does not guarantee that one of them is the tail domain. In addition, how can we ensure that the features stored in memory include those from the tail domain?

5   The necessary ablation study is missing. For example, the authors should conduct experiments to evaluate the effectiveness of the proposed method under the cases of Eq. (1) vs. Eq. (7) and Eq. (8) vs. Eq. (12).

6  The incremental improvement. In Tables 1-5, we observe that the proposed method achieves only limited performance improvement compared to the performance of the baseline.

7 Wrong template. The authors need to be more attentive to their submissions.

[A] Adaptive Hierarchical Representation Learning for Long-Tailed Object Detection, CVPR, 2022.

**Suitability:**

2

---

### Official Review · Reviewer_8oiJ · 2024-05-05

**Rating:** 5
**Confidence:** 3

**Summary:**

This manuscript proposed a method to solve the imbalance issue in Remote Physiological Measurement (RPM). Two imbalance issues were identified. For videos with the same physiological label, the domain can vary due to environmental setup. To solve this problem, the authors proposed anti-spurious domain center learning to learn domain-balanced embedding space. Another issue is related to the imbalanced number of video samples for different physiological labels. The authors proposed to solve it by adopting compact-aware continuity regularization to estimate the weights for hard and simple samples.

**Strengths:**

Overall, the paper was well written, with detailed method descriptions and comprehensive experiments.

**Limitations:**

However, there are some concerns that need to be addressed:
1. The coherence can be improved. For example, the term “memorybank” was introduced in section 3.2.1 but was used in section 3.1.2; At line 227, what is compactness?
2. In section 3.1.2: Due to the domain imbalance, the domain embeddings calculated for head, mid, and tail might also suffer from the imbalance issue. Using a clustering algorithm in Equation (3) to classify embeddings can be biased. Therefore, would it be better to add weighting parameters in Equation (6) to avoid this imbalance?
3. Typos: At line 579, “continuum”; At line 694, “inters”.
4. The discussion about experiment results is insufficient. Try to adjust your structure to allow more space for the discussion.
5. There is no discussion about limitations.

**Suitability:**

3

---

### Official Review · Reviewer_jvr1 · 2024-05-20

**Rating:** 4
**Confidence:** 2

**Summary:**

This paper presents a solution to the multi-level imbalance problem in the rPPG (Remote Optical Pulse Estimation) task. Two algorithms have been developed to address domain and physiological label imbalance issues.

**Strengths:**

Specifically, the authors alleviated the domain imbalance problem without providing any domain information. problem. Five datasets were used for experiments, and the experiments were relatively rich.

**Limitations:**

1. The ADL method proposed in this article employs unsupervised clustering to estimate the domain distribution of each physiological label sample. How can the authors prove that the clustering classification result is for the domain rather than the category? Additionally, can you compare the results of supervised clustering?
2. The CCR method proposed in this article evaluates the compactness of a category's feature distribution by calculating its embedding compactness and gives greater weight to more compact features. The author argues that this can balance the differences in importance between different categories in the data. However, a compact feature distribution usually leads to better classification. Why doesn't the model consider clusters with dispersed distribution?
3. As shown in Figure 1, domain imbalance and phys-label imbalance exist simultaneously. This article proposes two solutions to these two problems. However, in reality, both problems exist simultaneously. Is it possible to do the corresponding experiment?
4. The lack of feature visualization of the effects of each module.
5. Lack of reproducibility. I hope the authors can open-source the code for the article.

**Suitability:**

1

---

### Official Review · Reviewer_88ia · 2024-05-21

**Rating:** 5
**Confidence:** 3

**Summary:**

This study aimed to address the long-tail problem of physiological signal data among various datasets. In order to do so, this study designed two algorithms, ADL and CCR, for imbalance in rPPG task.

**Strengths:**

1. The manuscript is easy to understand.

2.  The experimental results are solid and promising.

**Limitations:**

1. The proposed framework include many hyperparameters (Eq2, 5, 9, 14 etc), how did the authors select these parameters? A more detailed explanation is strongly needed to support the results. In addition, some recent studies [1-3] are encouraged to compare and discuss with the proposed method.

[1] Dong, Y., Yang, G., & Yin, Y. (2022). Drnet: Decomposition and reconstruction network for remote physiological measurement. arXiv preprint arXiv:2206.05687.
[2] Yu, Z., Shen, Y., Shi, J., Zhao, H., Cui, Y., Zhang, J., ... & Zhao, G. (2023). Physformer++: Facial video-based physiological measurement with slowfast temporal difference transformer. International Journal of Computer Vision, 131(6), 1307-1330.
[3] Choi, J. H., Kang, K. B., & Kim, K. T. (2024, March). Fusion-Vital: Video-RF Fusion Transformer for Advanced Remote Physiological Measurement. In Proceedings of the AAAI Conference on Artificial Intelligence (Vol. 38, No. 2, pp. 1344-1352).

2. Format issue. The manuscript include many redundant information while appendix has more valuable experimental results in the current format. For instance, Figure 1 and 4-6 can be redesigned to save more space for experimental results. Motivation examples in section 3.1 and 3.2 can be moved to appendix to further make the manuscript more concise. In addition, why y2n = y2n-1=y2n mentioned in line 566? Also, what does the M represented in line 392 compared to the one in equation 10?  Moreover, there are some inconsistency between annotation font, such as L1 loss in line 641 and 646.

**Suitability:**

2

---

### Meta-Review · Area_Chair_bjKG · 2024-07-01

**Recommendation:** Accept (Poster)
**Confidence:** 4

**Metareview:**

This meta-review summarizes the evaluations from reviewers 88ia, jvr1, 8oiJ, and bJTX for submission 2082, titled "rPPG-HiBa: Hierarchical Balanced Framework for Remote Physiological Measurement." Overall, the reviewers lean towards accepting the paper for a poster presentation, with some requesting minor revisions for clarity and technical details.

**Strengths:**

* The paper proposes a novel method (rPPG-HiBa) to address the long-tail problem in physiological signal data caused by domain and physiological label imbalance. (88ia, jvr1, 8oiJ)
* The manuscript is well-written and easy to understand. (88ia)
* The experimental results are solid and promising. (88ia)
* The authors conducted comprehensive experiments to evaluate the proposed method. (bJTX)

**Weaknesses:**

* The framework description and figure can be improved for clarity. Reviewers requested clarification on specific notations (L_ASDCL, L_CACR, alpha) and the purpose of the estimator and auxiliary estimator. (bJTX)
* More details are needed on hyperparameter selection and justification. (88ia)
* The novelty of the approach, particularly regarding Anti-Spurious Domain Center Learning, needs to be better substantiated by addressing similarities to existing work (reference [A] mentioned by reviewer bJTX). (bJTX)
* The discussion on limitations and the effectiveness of each module could be improved. (8oiJ)
* Some reviewers pointed out minor typos and inconsistencies in symbol usage throughout the manuscript. (8oiJ, bJTX)

Based on the reviews, I recommend accepting submission 2082 for a poster presentation.